# Initial Testing of Robotic Exoskeleton Hand Device for Stroke Rehabilitation

**DOI:** 10.3390/s23146339

**Published:** 2023-07-12

**Authors:** Rami Alhamad, Nitin Seth, Hussein A. Abdullah

**Affiliations:** School of Engineering, University of Guelph, Guelph, ON N1G 2W1, Canada; rami@ramihmd.com (R.A.); sethn@uoguelph.ca (N.S.)

**Keywords:** rehabilitation robotics, hand rehabilitation, exoskeleton device, stroke rehabilitation, linear actuator

## Abstract

The preliminary test results of a novel robotic hand rehabilitation device aimed at treatment for the loss of motor abilities in the fingers and thumb due to stroke are presented. This device has been developed in collaboration with physiotherapists who regularly treat individuals who have suffered from a stroke. The device was tested on healthy adults to ensure comfort, user accessibility, and repeatability for various hand sizes in preparation for obtaining permission from regulatory bodies and implementing the design in a full clinical trial. Trials were conducted with 52 healthy individuals ranging in age from 19 to 93 with an average age of 58. A comfort survey and force data ANOVA were performed to measure hand motions and ensure the repeatability and accessibility of the system. Readings from the force sensor (*p* < 0.05) showed no significant difference between repetitions for each participant. All subjects considered the device comfortable. The device scored a mean comfort value of 8.5/10 on all comfort surveys and received the approval of all physiotherapists involved. The device has satisfied all design specifications, and the positive results of the participants suggest that it can be considered safe and reliable. It can therefore be moved forward for clinical trials with post-stroke users.

## 1. Introduction

Stroke is one of the leading causes of death and disability among adults worldwide [1], resulting in up to 50% of survivors having post-stroke disabilities [2]. The impairment or loss of upper limb functionality is one of the most common consequences of stroke. Upper limb impairment is a prominent issue that often prevents individuals from independently completing activities of daily living (ADLs). ADLs describe the fundamental daily activities that individuals can perform to maintain their quality of life. Rehabilitation is vital in treating post-stroke disabilities [3]. It involves a multidisciplinary team that can include doctors, physiotherapists, and healthcare assistants who play a vital role in the post-stroke rehabilitation process. Therefore, more demand for healthcare professionals and intervention therapy leads to higher healthcare costs.

New technical devices, such as rehabilitation robotics, have the potential to provide assistance with rehabilitating injured upper limbs through repetitive and task-specific therapy. The literature reported several reviews [4,5,6] showed positive upper limb motor function recovery results after using robotic rehabilitation devices for their therapy. Limited medical resources have created the need for rehabilitation robotics and other alternative therapy delivery methods. This situation has led to economic pressures to shorten the treatment periods of patients in hospitals [7,8]. Therefore, the role of robotic devices in rehabilitation research is becoming essential for providing rehabilitation treatment to individuals with stroke. The global aging of the population and an increasing number of stroke incidents point to such devices becoming necessary [9,10], with a special demand for therapy to regain lost arm and hand motor abilities. Robotic systems with precise and repeatable motions can fill some of these needs by providing intervention therapy that meets the established specifications of physiotherapists [11].

A fundamental approach for the rehabilitation treatment of impaired motor skills is task-oriented repetitive movements that help patients recover functionality [12,13]. A clinician can use robotic devices to help patients perform appropriate movements so that neurons in the brain are stimulated to reorganize and form new connections. The use of this brain characteristic, known as neuroplasticity, can lead to motor recovery.

Using techniques similar to those of physiotherapists, robots can deliver hand rehabilitation in both passive and semi-active modes. For passive treatments, the robot’s motion is the sole driving force that moves an individual’s fingers. Conversely, semi-active treatment involves using the volitional power of the individual’s own fingers and hand to drive the motion with some assistance from the robot. Therefore, treatments can mimic those of a physiotherapist, either by directly manipulating the motion of the fingers and hand or acting as a guide.

In addition to facilitating practiced movements, robotics technology provides the means of collecting quantifiable data to monitor the patient’s performance [11]. Physiotherapists and patients may be better able to observe any minor changes and improvements in the quantifiable signs of progress, their outcomes, and the treatment efficiency [14].

The development of robotic systems for rehabilitation settings has followed several paths. Initial clinical studies in rehabilitation robotics were performed using the MIT-MANUS robotic arm developed by MIT to investigate the effectiveness of target-based robotic-assisted movements [7]. However, many studies related to the post-stroke robotic rehabilitation of the upper limb, such as those involving MIT-MANUS [15] and GENTLE/S [16], focused primarily on arm joints, including the shoulder and elbow, rather than the hand and the finger joints [3]. As a result, the therapeutic exercises for some of these devices did not include the finger movements necessary to achieve pinching and grasping. These movements are required to obtain functional improvement, which is necessary for performing ADLs independently [17,18,19].

### Robotic Rehabilitation for Hands

Several studies on the upper limb joints found that improvements occurred primarily in the shoulder and elbow [20,21]. Because the joints exercised were the only joints to show improvement, the development of hand devices became necessary. For hands, the greatest need was to develop support for performing functional tasks, specifically the pinch and grasp motions. These motions allow for the manipulation of objects, which is necessary to achieve activities of daily living. As a result, several new forms of hand rehabilitation devices using robot-aided therapy have been researched in recent years. These devices can be categorized into two main types of robotic systems: exoskeletons and end effectors [22].

Exoskeleton-Type Systems

These systems are mounted or worn by the patient on their hand. The patient controls its motion directly [23,24]. The Hand-Wrist Assisting Robotic Device (HWARD), developed at the University of California, is a standalone device for treating stroke patients. The HWARD system is considered one of the first exoskeleton devices that provides actuation on the thumb plus the fingers grouped as a whole. It allows for manipulating household objects during training in a virtual world [25]. Another example of an exoskeleton device is the PMHand, developed by McConnell et al. at Heriot-Watt University [26]. It uses a 3D-printed exoskeleton frame that utilizes wires linked to a motor to generate the hand’s and its fingers’ movement. An alternative to this is found in the emergence of a new version of wearable devices known as soft robots. These devices are fabricated from easily deformable materials, such as gel or soft polymers [27], and tend to be less complex, safer, more portable, and flexible.

End-Effector-Type Systems

These systems interact with the patient through a single point and are either attached to the hand of the patient or gripped by the hand [23,24]. One first end-effector device developed for hand and wrist rehabilitation was the Space Interface Device for Artificial Reality (SPIDAR). This device was invented in 1989 at the Tokyo Institute of Technology [28]. Another approach to hand and wrist rehabilitation consists of a knob-like two degrees of freedom interface for the user to grasp and use to exercise forearm movement. Another example is the haptic knob, which was developed by the National University of Singapore [29].

Moggio et al. published a meta-analysis comparing exoskeleton versus end-effector robot-assisted therapy for finger-hand motor recovery in stroke survivors [30]. Aggogeri et al. [31] and Kabir et al. [32] reviewed the research status in hand rehabilitation robotic technology and evaluated several devices to describe the current state of the art [31]. Another overview of hand rehabilitation robotics, which addressed everything from the hardware systems to the training paradigms, was published in 2017 [33].

Exoskeleton devices provide patients with active and passive rehabilitation using different sources of actuation and mechanisms. They both describe the means used in devices to generate and translate power from actuators to move the fingers. Actuation can be classified as elector motor, hydraulic, and pneumatic [33]. Electromotors are widely used for hand rehabilitation design because they are reliable and easy to install and control with high precision [34]. In contrast, hydraulic and pneumatic actuators are less used due to the issues related to noise, leakage, control, and storage of the compressed air and fluid. They also depend on multiple components, including the compressor, different values and regulators, pump, and control elements, which add complexity and cost to the control and design of the rehabilitation device [35].

The actuation mechanism’s main function is to transform the motion of the actuator to the fingers to achieve the hand therapy exercises. They depend on the nature of the actuator and can be categorized into main classifications: pneumatic actuation [27,36], linkage-based actuation [37,38,39,40,41], cable-driven systems [33,42,43,44,45,46,47,48], and gear-motor actuation [32]. Linkage and gear-motor actuation are commonly used for hand rehabilitation devices. However, they had several rigid linkages, pullies, gears, and mechanisms to control the finger motions and their joint. As a result, bulky and complicated exoskeleton hand devices were designed and developed that are costly, heavy, and caused uncomfortable experiences for the users and had limited workspace area. All of these affected the device’s biomimetic qualities, and some devices resulted in misalignment with the finger’s anatomic axis during motion [27,33]. On the other hand, soft exoskeleton hand rehabilitation devices are simple, not rigid, and lightweight. ExHand Exoskeleton is a fabric-based soft hand exoskeleton for assistance in ADLs developed and evaluated by ten healthy users using a glove weighing 137 g [49]. However, they were mainly actuated by the pneumatic system, which requires complex components and control systems and has low accuracy [27,50,51,52,53].

The cable was also frequently used as the actuation mechanism in hand rehabilitation robots [42,43]. Some devices used a soft glove for hand rehabilitation with a cable-driven mechanism [54,55]. Some popular devices are the Exo-Glove [56], HandCARE [57], and Gloreha [58]. A cable-driven mechanism attached to a soft glove instead of the rigid links can be an alternative to rigid linkage devices [46,54,55]. Soft devices driven using cable mechanisms tend to be lighter, more comfortable, low-cost, and portable as the motors and sensors can be placed away from the hand device attachment or glove. However, using gloves can have limitations due to the different hand sizes and could induce significant joint reaction forces [34,43].

The common theme observed in the hand device studies presented in the literature is the inclusion of sensors to measure the position of the patient’s fingers and thumb, the force produced by them, or both. In addition, accomplishing the reduced power of actuation is an important goal to reduce the cost and complexity of the design. When looking at how each system actuates the fingers, it becomes apparent that the 21 DOFs available in the fingers of a human hand are simply too many to actuate and exercise individually. It would affect the system’s complexity and weight, making some of these devices relatively expensive to acquire and maintain. Another theme was hand rehabilitation focused on ADL tasks in 3D space, which is considered the most practical and beneficial to patients [29,59,60,61]. These tasks involve training the patient on repetitive motion patterns, such as gripping and releasing an apple.

The proposed design approach addressed some of the challenges reported in the literature. It focused on developing a hand rehabilitation device with a flexible structure supporting different hand, finger, and thumb sizes. Provide a light and secure link to the auction and sensing system that was placed away from the palmar side of the hand so that the hand can be free to interact with real-world objects. This feature was selected based on our discussions with physiotherapists. The free hand helped users undergo training involving repetitive motion patterns with real objects. The proposed device features training for functional tasks with the four fingers grouped together in a reaching motion and the thumb actuated separately. Most other systems reported in the literature actuate each finger individually. The device used in the current approach simplifies the actuation, sensing, and control complexity because it uses only two motors rather than five. One motor activates the four fingers as a group, and the second activates the thumb. Moreover, the proposed hand device is light, with the total hand attachment at a weight of 246 g. It can be considered portable compared to other devices reported in the literature, making it comfortable for patients to wear and use. These features help to differentiate the proposed device from most of the systems reported in the literature, which actuate each finger individually. As a result, the specifications of the proposed device reduce its total complexity and cost.

This paper presents the results and analysis of a pilot study that examined the accessibility and reliability of the device, as well as how comfortable it was to use for healthy adults. These essential elements would affect stroke patients’ acceptance of the device and its efficacy. A pilot study that involved 52 healthy participants was conducted as a first step to obtaining regulatory permission for a clinical trial involving stroke patients. The study provided feedback regarding the comfort of the device. The quantifiable data collected for the system produced similar repeatable results for the repetitive motions being performed. Repeatability is the ability of the device to produce similar, quantifiable results as well as to operate predictably and reliably; this is critical when designing a rehabilitation device.

## 2. Materials and Methods

### 2.1. Device Criteria and Constraints

This device was the culmination of multiple consultations with physiotherapists as well as those of potential study participants contacted during the initial design process. The feedback from the physiotherapist and the stroke patients (users) is a critical component in developing a robotic rehabilitation device [11]. Design components were created based on Health Canada Safety Regulations as well as key criteria noted by physiotherapists. These criteria are summarized in Table 1 below and were fully incorporated into the design.

The specifications of these physiotherapists were based on their 10+ years of experience in treating stroke patients for the recovery of hand mobility. The device was intended for use with subjects with limited mobility in their hands and at approximately Stroke Stage 2 or 3 on the Chedoke–McMaster Stroke Assessment Scale (CMSA) [35]. Subjects in Stage 1 who could not use their hands and those who were hypertonic could not use the device. User feedback was collected to assess the adequacy of comfort and fit and to determine any possible need for major design revisions that might be required in response to complaints or suggestions.

The average and maximum forces necessary to open the hand of a stroke patient were determined by tests conducted at the Regional Rehabilitation Center, Hamilton Health Sciences in Hamilton. The test included eight stroke patients in determining the forces required to extend the fingers of their impaired hand. The results of this test are included in Table 2.

### 2.2. Design of the System

The system’s design is based on cables connected to two linear actuators that deliver the force necessary to open the hand of the user. The actuators are meant to be used with exercises to open only the hand, which was the main concern expressed by the physiotherapists. Since the system uses cables to actuate the hand, it could not be made backdrivable. While this feature is important for safety in geared systems, there are two reasons it could be omitted from this design. First, the system is used by an operator who has access to the emergency stop button and constantly observes the patient. The cables are also attached to the hand utilizing a comfortably padded hand brace (Figure 1) that has a quick-release latch that can be removed easily in an emergency. Cables run from the brace to the box containing the two linear actuators and sensors. This box is placed behind the user on the same side of the treated hand. The load sensor for each actuator sends the force reading back to the device controller and the connected personal computer (PC).

The actuators sat behind the person using the device and opened the user’s hand. The load sensors and actuators were also connected to a desktop computer that runs control software that allows a physiotherapist to manage the system.

The three main components to secure the fingers and thumb are shown in Figure 1. The components were designed to be ambidextrous, accommodate a wide variety of hand sizes, and be easy to place on the hand. The tips of the fingers were grouped together and attached via a comfortable elastic tip strap. This strap was then connected to the actuators in the system via a cable. The physiotherapist can adjust where the strap is connected along the cable to accommodate different hand sizes. A larger hand would have its fingers attached to the end of the cable, whereas a shorter hand would be attached higher up. A thumb tip strap, seen in Figure 1, was placed onto the user’s thumb and connected to the second linear actuator via another cable. This cable was attached to the hand brace with a wrist strap to prevent finger hyperextension.

A mid-link strap, seen in Figure 1, was attached to the hand brace with a wrist strap to prevent finger hyperextension. The mid-link strap was also used to prevent the cable from contacting the hand of the user while keeping the cable connected to the actuators and tip straps. Similar to the mid-link strap, the wrist strap was also designed to catch the mid-link strap and prevent wrist hyperextension. These components were all stress-tested using SolidWorks, a 3D modeling package. Forces approximating those that the components might experience during use were applied through a simulation. The components did not break or significantly deform under the forces applied.

The tip, mid, and wrist straps were designed and tested in SolidWorks before production using the 3D Printer. The first step of analyzing the components was determining the material properties SolidWorks required. Once the material properties were prepared, testing began on the tip bottom piece section, mid-section, and wrist section to ensure that all components would maintain structural rigidity if the user and linear actuators applied a reasonable force. Testing at the Regional Rehabilitation Center, Hamilton Health Sciences in Hamilton, Table 2, proved that the maximum force required to extend a stroke patient’s palm is 25.48 N. In order to ensure optimum safety, a Factor of Safety of two was used, and the parts were exposed to a force of 50 N along anticipated points of stress to examine their reaction to such a force. The tip strap consists of two components (top and bottom), and the bottom piece is attached from both ends to the top piece, which is attached to the cable; while the cable is being pulled, the bottom piece will experience a force across its top surface. This force was modeled in SolidWorks, and the results indicated that the maximum expected displacement is 7.94267 e-006 mm, which is well within the tolerance of design specifications, Figure 2.

The mid-section also proved that it could handle the forces it is expected to experience during use; the maximum displacement expected is 4.1607 e-003 mm. The wrist section piece was also similarly tested, and the maximum displacement anticipated is 3.2705e-003 mm, Figure 3.

All the components except the hand straps were manufactured using a 3D printer at the School of Engineering. The combined mass of the components that were manufactured was 246 g. The measured mass of each component is presented in Table 3.

A PC with a software subsystem communicated with the load sensors that were used to measure the force of the actuator through a Texas Instruments data acquisition (DAQ) board. The DAQ sent control signals to the control circuit of the actuator, and each actuator communicated to the software subsystem via the same DAQ. Linear actuators were used to deliver the force to open the hands of the subject. The system used linear actuation to keep the design simple and easy to maintain.

Given the need for an average force of approximately 15 N and a maximum of approximately 25 N, two Firgelli FA-PO-35-12 linear actuators were selected for their ability to meet the force requirements at a reasonable cost. The actuators had a stroke length of 152 mm to ensure sufficient motion to open a variety of hand sizes.

Two Loadstar iLoad TR load sensors were selected for the hand device. They were attached directly to the actuators to measure the force exerted by both actuators on the user’s thumb and fingers. The sensors can measure forces up to 222 N at a maximum sample rate of 150 per second. These sensors were connected directly to the personal computer being used for control via a USB port. The sensors and the cable used to actuate the hand were attached with a welded bolt. All these components were housed in a fireproof PVC box with an emergency stop button that can be used to shut down the power to the two actuators.

A software system was developed to control the linear actuators, read the load sensor values, record user data, and conduct hand training exercises. The software also controlled the system’s operating mode which is described in Section 2.3.

### 2.3. Passive and Semi-Active Modes

The two training exercise modes available are the passive mode and the semi-active mode. Both modes direct exercises that repeatedly open and release the user’s hand at varying activity levels. In the passive mode, the device emulates the extension of the hand during the initial training sessions. A chart showing the flow of control for the software is shown in Figure 4. The limit to which the fingers and thumb can be extended is set by the physiotherapist, who sets the system to pull on the user’s fingers/thumb until they are fully extended. The result of this action is considered the full extension position for each user. Once this limit is achieved, the actuators release the tension on the cables and allow the user to return their fingers/thumb to the initial position in one of three ways: (1) by means of the user’s own volition, (2) by use of the muscle capacity of the user, or (3) with the assistance of the physiotherapist. The actuators then tense and release the cables for the number of repetitions determined by the physiotherapist. Visual and audio prompts let the user know when the device is moving before the move is executed.

The semi-active mode, shown in Figure 5, allows for greater interaction between the device and the participant. Following an on-screen prompt, the user is expected to open his/her hand independently, with the device assisting only if necessary. If the hand is not moved to the correct position or cannot be moved at all, the system initiates actuation and moves the user’s hand to the desired location. The tension of the cable indicates whether the fingers/thumb have been moved. If a drop in tension occurs, it is assumed that the hand has been extended. If there is no change, the hand is deemed not to have been moved, and the system is expected to help the user complete the movement.

### 2.4. Experimental Goals

The goal of the pilot study was to determine if the design specifications for the system had been met and if the device was suitable for a clinical trial. The analysis of this paper focuses on an evaluation of the comfort, accessibility, and repeatability of the device. Repeatability is an important aspect to develop for a rehabilitation device as it leads to predictable or reliable behavior by the system. This reliability is an important safety issue as it ensures that the system only moves within a comfortable and acceptable range of motion. This range of motion will be established by a physiotherapist for each patient.

In order to ensure that the performance of the device was repeatable and accurate, a statistical analysis of the pilot study data was carried out. The repeatability tests analyzed the force values that corresponded to fully extending the fingers of each user. This analysis was carried out on the data to ensure that there was very little variation between the peak force values of the cycles in each exercise.

The comfort level of the participants was investigated to ensure that those using the device had a positive experience. After using the device, a feedback survey designed to gauge the comfort of the participants was issued to each subject in the trial. Developed in consultation with both physiotherapists and statisticians, the survey produced feedback that addressed three aspects of the device—comfort feedback, hand size poll, and overall comments. The comfort feedback looked at the subject’s level of comfort during both passive and semi-active exercises. The hand size component sought to determine how well the device fit the user. This information, along with the participant’s personal information, was used to examine the limits of the size of the hand comfortably accommodated by the device. The experimental methodology was approved by the University of Guelph Research Ethics Board prior to the start of the study (protocol no. 10FE010). All interested participants were presented with information about the experiment and then signed a consent form if they agreed to participate.

### 2.5. Methods

Testing was carried out on 52 age-appropriate healthy adults grouped by age: under 50, 50–59, 60–69, and over 70. The distribution of the individuals is shown in Figure 6. On the feedback survey forms issued, 10-point Likert scales were used to determine the levels of comfort experienced by individuals for the finger and thumb components of both modes.

All participants started with their hands forming a fist and were instructed to relax their hands so that the system could passively lead them. The repetition began with the linear actuators moving to a distance set by the investigator to cause tension in the cables. This cable tension caused the participants to be passively led into the fully extended finger and thumb position.

Each participant underwent nine extension motions using the system to ensure that each session with the device produced similar force readings as they were passively led from full finger and thumb flexion to full extension. The cable tension was released at the end of a single repetition; at this point, the participant would reform a fist for the next repetition. Testing was also carried out in a semi-active mode, in which participants would open their hands partially without being passively led. Once the system detected that the hand was partially open, the robotic system would fully open the hand.

Another point of analysis was the time required to attach the device to the hand of the participant. According to clinicians, this is an important point, as a device that is difficult to prepare or requires a long set-up time is considered undesirable.

## 3. Results

The results collected from the participants through the Likert scales and their standard deviation (SD) are presented in Table 4 for the fingertip strap and in Table 5 for the thumb piece. The number of participants is noted in parentheses for each age group.

The device was found to be comfortable, with no significant differences in the comfort scores between the finger and thumb pieces. The mean of each of the four Likert scales was approximately 8.5 out of 10. The standard deviation of each of these scales was approximately 1.4, indicating a strong consensus amongst participants that the device is comfortable.

A Kruskal–Wallis test was performed to adjust for the non-parametric nature of the data collected on the Likert scale. The test determined that age, gender, and weight had no significant effect on the ratings for the device. Another measure of comfort considered was the time required for participants to become used to the device. Participants were asked to indicate by which repetition they felt sufficiently accustomed to the motion of the device to be comfortable moving along with its motion. The mean number of repetitions it took to adjust to the device was 1.58 in the passive mode and 1.52 in the semi-active mode. The full results of these tests for both the passive and semi-active modes can be seen in Figure 7.

One of the main goals of the design was to make the hand device practical to take on and off regularly and easily. The time required to place the glove on each participant’s hand was recorded and listed in Table 6. The average time required to place the glove on the hand was 56 s. No significant differences were detected across the various age and gender groups for the time required to place the device on the user’s hand.

The device was found to be repeatable in terms of intrapersonal reliability. An example of this is the load sensor data gathered during a passive test of the linear actuators, which are retracted to extend the hand of each participant. These data are shown in Figure 8 (Bottom) for the fingers and Figure 8 (Top) for the thumb. In these figures, the position and force are plotted against the time. The position value is the distance at which the actuator moved. The force values are the amount of tension in the cable that was used to open the hand. In all the figures shown, it is observed that the peak force gradually declines. The first iteration is likely the participant’s first exposure to the system, so the individual is not fully used to the motion of the device and is likely to react with greater resistance (limit test cycle). Subsequent repetitions allow the participant to become used to the motion of the system. As the participant learns to accommodate to the device, this produces slightly lower peak force values. Figure 8 plots the passive mode test, which was repeated for nine cycles (one limit test cycle and eight exercise cycles). Therefore, an analysis of the results was completed on the eighth exercise cycles only.

A one-way ANOVA was used to detect whether the differences between the forces produced when participants repeatedly opened their fingers demonstrated a sufficiently small variation. Peak force values of each between-group variance were significantly higher than the within-group variance. The variance was expected, and the results show that a large variance between different participants (users) was present since the force required to open the hand heavily depended on the overall weight, muscle strength, and size of the user’s hand. A repeatable device would demonstrate that each participant would have little variance in their peak readings. The within-user variance, which is essentially the variance between the peak samples from each user, was sufficiently small to warrant a positive conclusion on the repeatability issue. The results of the ANOVA test for the finger and thumb samples are listed in Table 7.

It can be concluded that the system is repeatable in its output, and the performance of both the finger and thumb actuation is comparable.

## 4. Discussion

This study investigated design issues related to the safety, comfort, and repeatability of a cable-driven hand exoskeleton device to actuate the hand during the treatment of stroke patients. The system needs to be put on and taken off the patient quickly and easily and be comfortable for the person receiving treatment. A hand device must also be capable of reliably providing repeatable motions comparable to those used in conventional therapy treatments. Any readings obtained by the system must also be repeatable to ensure that the system will operate predictably and remain safe.

Unlike most conventional hand devices, the tested system actuates all four fingers, which are separated from the thumb and grouped together at the back of the hand to regain grasping abilities. This approach reduces the number of actuators and components needed to control both the operation and the system. This method also reduces the setup time to 56 s, which can be considered sufficiently rapid to be convenient for physiotherapists under strict time constraints. The time needed to place the device on the hand was further reduced once the physiotherapists became more familiar with the device and used it repeatedly.

Participants reported no fatigue and were comfortable; thus, a modular lightweight exoskeleton system can be used effectively without discomfort. It was found to operate reliably, safely, and easily and was comfortable for healthy individuals. From this, it can be expected that any pain or discomfort experienced by the user will be exercise-related and not induced by the device.

The study found that the range of force values needed to open the hand of the users had a great amount of variance across all individuals due to different hand sizes, weights, and strengths. The results of the analyses found that although the hand forces demonstrated a relatively large variance, the within-user readings were found to be consistent. From this, physiotherapists and clinicians using the device can be assured that the motions being provided are repeatable. They may also be able to actively monitor force readings over time to observe changes in the amount of force needed to open a hand. This force information can be used to quantify the treatment progress.

It is anticipated that physiotherapists can use robotic systems like this one to treat individuals by using repetitive motion exercises. At the same time, physiotherapists can gather quantifiable data to gain more insight into the condition of their patients. This proposed system can provide the physiotherapist with a lightweight and simple tool to augment existing methods of therapy and evaluation.

## 5. Conclusions

This paper presents the initial testing of an exoskeleton hand device that provides passive and semi-active rehabilitation of fingers and thumb. The prosed design approach addressed some hand device challenges reported in the literature. Consequently, it focused on developing a hand rehabilitation device with a flexible structure supporting different hand, finger, and thumb sizes. A light and secure link is provided to the action and sensing system through a cable mechanism to position it away from the palmar side of the hand. Therefore, the user’s hand can be free to interact with real-world objects. This feature was selected based on our discussions with physiotherapists to help users undergo training involving repetitive motion patterns with real objects. The proposed device features training for functional tasks with the four fingers grouped together in a reaching motion and the thumb actuated separately. Most other systems reported in the literature actuate each finger individually. The proposed approach simplifies the device’s actuation, sensing, and control complexity because of the use of only two motors and sensors rather than five. The proposed hand attachment is light, with a total hand attachment weighing 246 g. It can be considered portable compared to other devices reported in the literature, which actuate each finger individually. These features help to differentiate the developed device from most of the systems reported in the literature.

The results of the device’s initial test are presented, and the device was found to be comfortable and easy to use by the participants. The measurements provided by the two load sensors were found to be repeatable and reliable. The device functioned well when administering a passive form of treatment. This study has determined that the device can provide acceptable and comfortable training through repetitive passive exercises when used by healthy adults. The device appears to produce reliable measurements and may now proceed to be tested in clinical trials with stroke subjects. The lightweight and modular nature of this system as a standalone may allow it to be integrated with existing systems that deliver treatments for upper-limb rehabilitation.

## Figures and Tables

**Figure 1 sensors-23-06339-f001:**
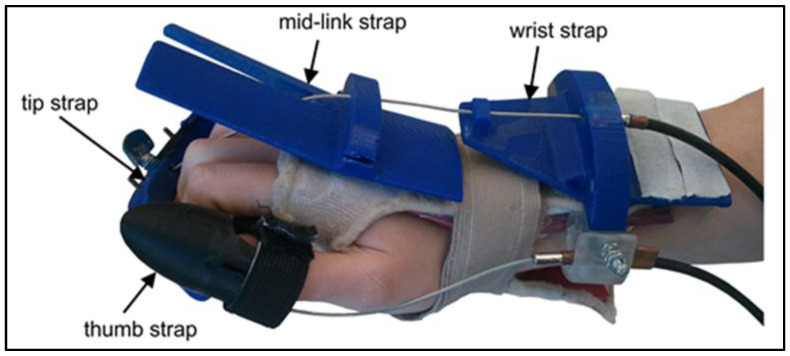
Final design overview.

**Figure 2 sensors-23-06339-f002:**
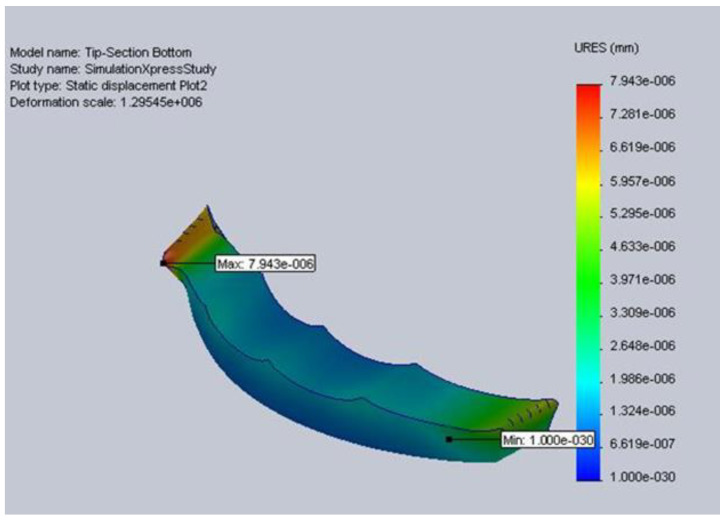
Tip section static displacement.

**Figure 3 sensors-23-06339-f003:**
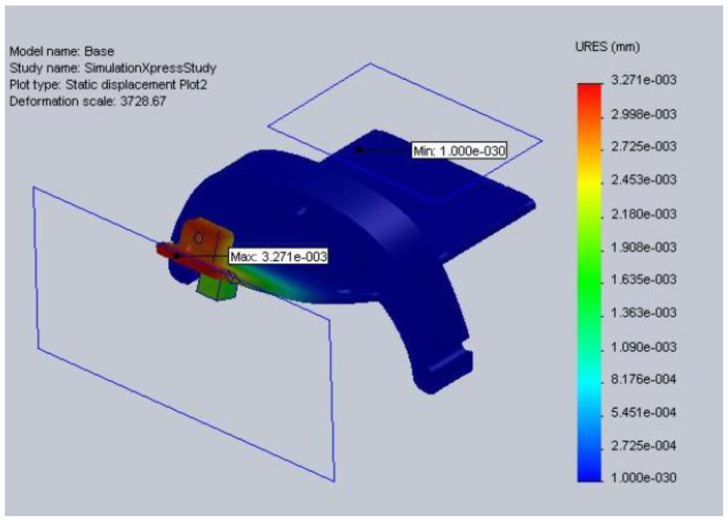
Wrist section tip static displacement.

**Figure 4 sensors-23-06339-f004:**
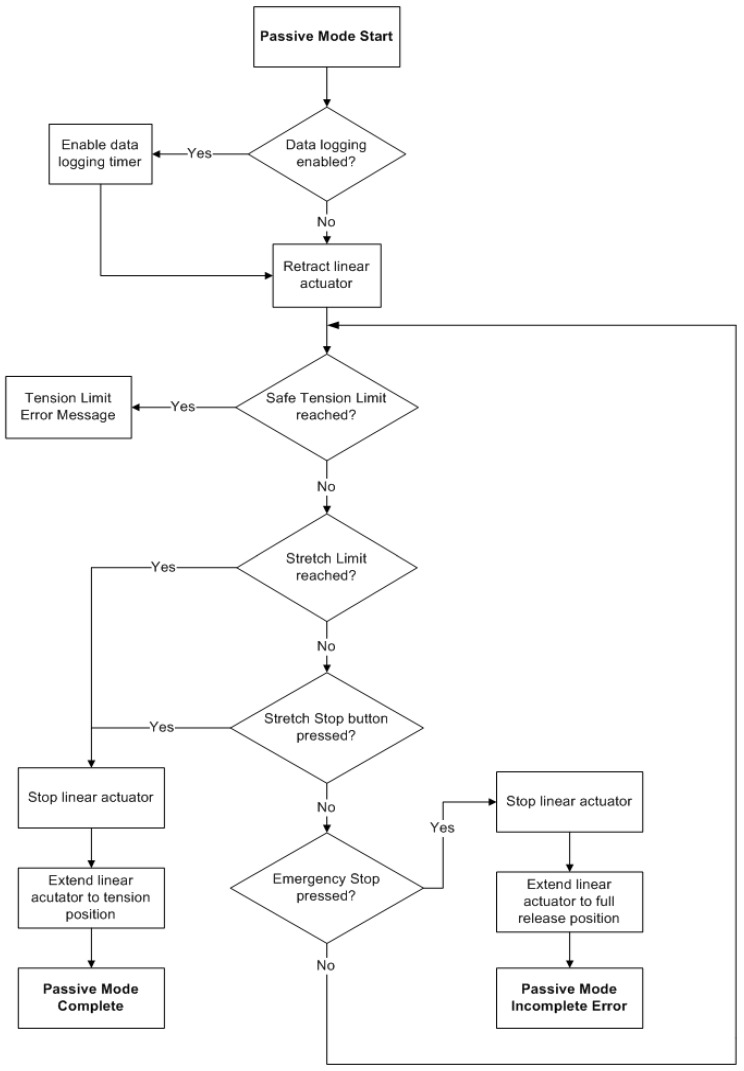
Flow chart describing the passive mode process.

**Figure 5 sensors-23-06339-f005:**
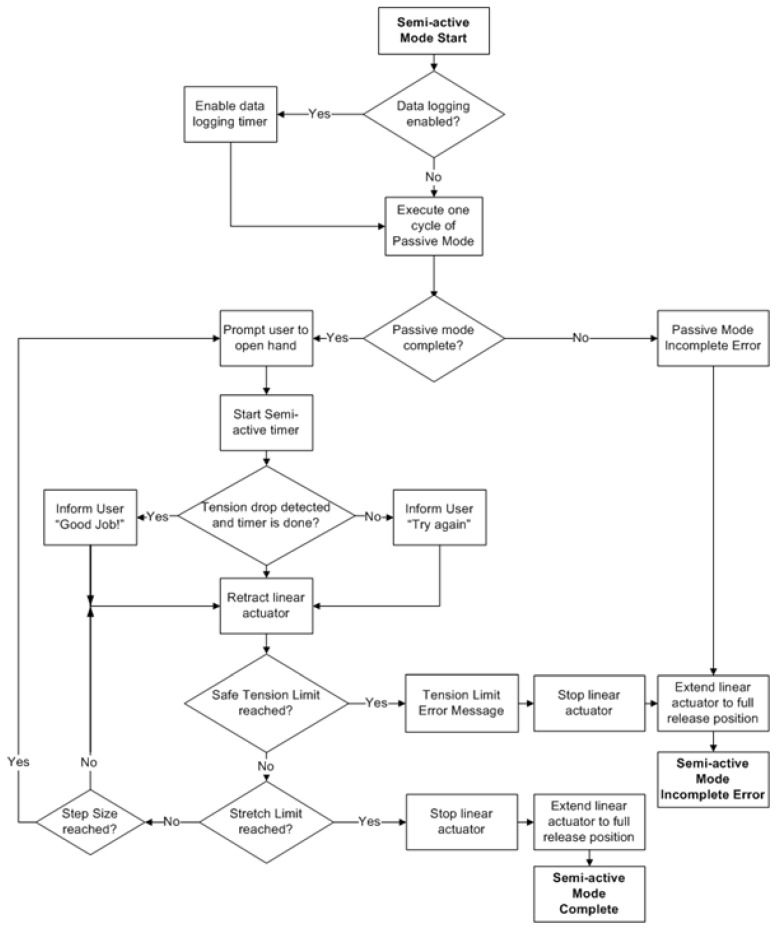
Flow chart describing the smi-active mode process.

**Figure 6 sensors-23-06339-f006:**
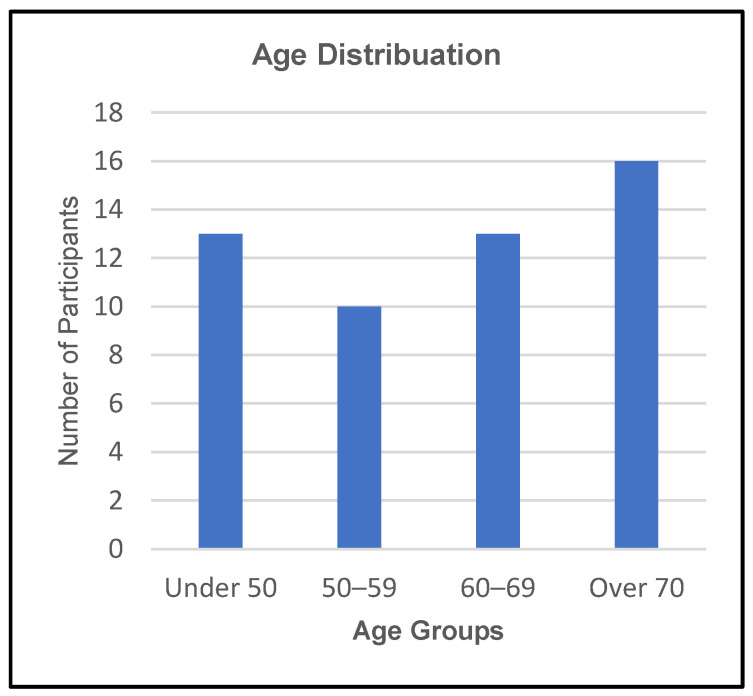
Total number of participants in each age demographic.

**Figure 7 sensors-23-06339-f007:**
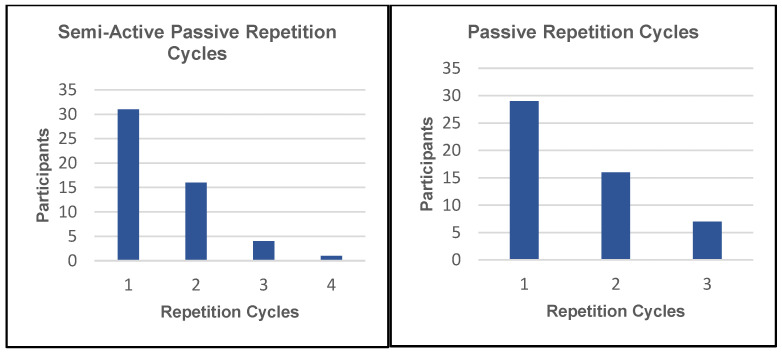
The number of repetition cycles required before participants became accustomed to the device in the passive mode (**right** graph) and semi-active mode (**left** graph).

**Figure 8 sensors-23-06339-f008:**
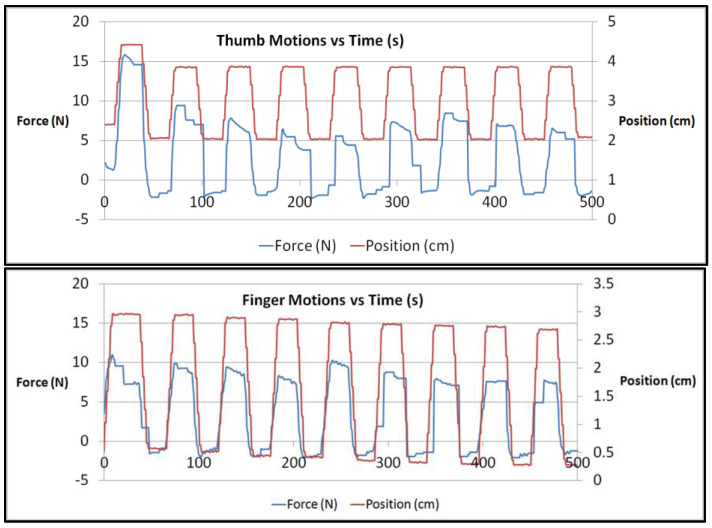
Force (N) and position (cm) against time (s) for nine cycles of passive finger (**Bottom**) and passive thumb (**Top**) motions.

**Table 1 sensors-23-06339-t001:** Summary of criteria from physiotherapists and users.

Physiotherapists Recommendations	Users Recommendations
- The device placed on the user’s hand needed to be less than 800 g in weight and less than 25 cm in length to accommodate various hand sizes while not fatiguing the individual. - The device needed to be able to withstand patient and gravitational forces. - The device needed to be able to actuate the wrist. - The device must be easy to remove or shut off, if necessary, in case of an emergency. - The device or system must be portable for easy transport around a hospital. - The design should protect against hyperextension of the fingers.	- The tip of the strap covering the fingers should be curved instead of flat for improved comfort. - All plastic components should not contact the skin for the purpose of comfort.

**Table 2 sensors-23-06339-t002:** The force required to extend the fingers of an impaired hand.

Patient	Trial 1	Trial 2	Trial 3	Average (N)	Maximum (N)
1	17.64	20.58	11.76	16.66	20.58
2	15.68	20.58	21.56	19.27	21.56
3	9.8	6.86	5.88	7.51	9.8
4	8.82	8.82	12.74	10.13	12.74
5	2.94	2.94	3.92	3.27	3.92
6	23.52	22.54	25.48	23.85	25.48
7	17.64	18.62	20.58	18.95	20.58
8	21.56	20.58	20.58	20.91	21.56
			Average	15.07	
			Maximum		25.48

**Table 3 sensors-23-06339-t003:** Measured mass of each component in the device.

Component	Mass (g)
Thumb Section	18
Tip Section	41
Mid-Section	54
Wrist Section	69
Brace	64
Total	246

**Table 4 sensors-23-06339-t004:** Finger comfort results for the passive and semi-active modes (Scale 0 to 10).

Age Group	Female (no.)	Male (no.)	Passive Average	Semi-Active Average
Under 50	7.2 (6)	8.6 (7)	7.9	7.85
50–59	9.4 (5)	8.6 (5)	9	9.2
60–69	8.5 (8)	8.8 (5)	8.65	8.9
70+	8.3 (10)	8 (6)	8.15	8.1
Average	8.35 (29)	8.5 (23)	8.43	8.51
SD	0.88	0.30	0.42	0.55

**Table 5 sensors-23-06339-t005:** Thumb comfort results for the passive and semi-active modes (Scale 0 to 10).

Age Group	Female (no.)	Male (no.)	Passive Average	Semi-Active Average
Under 50	6.8 (6)	8.4 (7)	7.6	7.4
50–59	9.8 (5)	9 (5)	9.4	9.5
60–69	8.8 (8)	9 (5)	8.9	8.75
70+	8.3 (10)	8.4 (6)	8.35	8.35
Average	8.43 (29)	8.7 (23)	8.56	8.5
SD	1.08	0.30	0.66	0.75

**Table 6 sensors-23-06339-t006:** The time (in seconds) participants require to put on the glove during the trial.

Age Group	Females	Males	Average
Under 50	52.7	56.1	54.4
50–59	49.3	58.9	54.1
60–69	58.6	60	59.3
Over 70	54.4	59	56.7
Average	53.7	58.5	56.1
SD	3.86	1.67	2.41

**Table 7 sensors-23-06339-t007:** ANOVA test results for repeatability testing of the hand device.

		Sum of Squares	df	Mean Square	F	Sig.
Finger Movement	Between Users	14,373.289	51	281.829	57.201	0.000
Within Users		1793.439	365	4.927		
Total		16,166.728	415			
Thumb Movement	Between Users	7099.213	51	139.200	55.702	0.000
Within Users		909.637	364	2.499		
Total		8008.85	415			

## Data Availability

The data supporting this study’s findings are available on request.

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
