# Peer review of "Initial Testing of Robotic Exoskeleton Hand Device for Stroke Rehabilitation"

_sensors, 2023, doi:10.3390/s23146339_

Round 1
Reviewer 1 Report
The authors presented an exoskeleton hand for post-stroke rehabilitation therapies. Tests of comfort and repeatability were conducted with 52 healthy individuals. The exoskeleton is actuated by two linear actuators and cables. In addition, the force is measured with two load sensors, and the whole system is integrated into a fire-proof PVC box.
In my opinion, the paper is well written. However, the presentation is very poor. Regarding novelty, it is not clear what is the contribution to the state of the art of the presented approach, since wire-actuated rehabilitation devices is not a new topic. The presented study seems more like preliminary work and needs improvements to achieve the technical quality of the Sensors Journal. The paper is not accepted in its current form. The authors are encouraged to enhance the paper and resubmit it.
Specific comments:
· Line 73 – life [17-19].;
· Line 80 - ADL was already defined in line 30. The acronyms are defined just once and the first time they appear;
· Line 106 and 108 – it is not common to include the year in et al. Please standardize the citation style;
· Line 159 – In Table 1 seems something is lost. “-All plastic components should not contact the skin for the purpose of”. Please verify and correct it;
· Same table – The device placed on the user's hand needed to be less than 800 gm in weight. Is it recommended to use g;
· Figures and Tables’ titles are without the period;
· Figure 2, does not bring any new information to the paper. It can be removed;
· All Figures have poor resolution. Please improve the resolution;
· To highlight the paper’s contribution, the authors must include a comparative table with similar research;
· In some parts of the introduction, there is no connection between sentences, as in line 31, which can cause the reader to lose the thread: “ADL describes the fundamental daily activities that individuals can perform to maintain their quality of life. Rehabilitation is vital in post-stroke treatment [3]”;
· Also, in the introduction it is mentioned that rehabilitation robotics has the potential to assist with rehabilitating injured upper limbs through repetitive and task-specific therapy. Therefore, mentioning some of the advantages of robotics would be interesting;
· In the paragraph starting on line 220, it is recommended to rewrite the paragraph because it first mentions the characteristics of the sensors before presenting the sensor;
· In Section 2.2 it is mentioned that the components were all stress-tested using SolidWorks, a 3-D modeling package. It would be interesting to see the results of the stress tests and if possible, compare them with similar devices;
· Within the recommendations of the therapists, it is mentioned that the device should be portable. And in the text, it is not mentioned if the device is portable. Please clarify if the device is portable or not, how can it be portable? On the other hand, it is mentioned that the device has a total weight of 246 g, which makes it a light device, but there is no mention of the total weight of the system, the weight of the components, and the box since the weight of the system should also be light to be easily transported by clinical centers;
· In Table 6, the caption is on another page, and the line number indicators are above the table. It may also be a formatting problem. Please check it;
· Regarding the results, it is recommended to present the standard deviation, since it can be information that can influence the analysis of results;
· In Figure 7 it is possible to place the graphs horizontally to save space;
· Figures 8 and 9 could be merged;
· Finally, in the discussion, it is recommended to perform a better comparison with similar devices, weight, exerted force, and what advantages/disadvantages present the control of four fingers as a group and the thumb apart. What difficulties were encountered and what is expected in future works?
The paper is well written. Minor spelling errors were find.
Reviewer 2 Report
The paper presents a novel robotic hand rehabilitation device aimed at treatment for the loss of motor abilities in the fingers and thumb due to stroke. The device used in this approach simplifies the complexity of the actuation and control because it uses only two motors.
Variants of such rehabilitation systems are known in literature. Hence the novelty of this work is quite limited. The constructive solution presented in the paper is a fairly simple one, its innovative character being questionable. The main deficiency of the proposed solution is, as asserted in line 166, that the rehabilitation equipment addresses solely the extension of the wrist, while flexion remains unsolved. Thus the equipment’s autonomy is diminished, as its utilization still requires the presence of a physical therapist. This is an issued that should be solved.
The paper layout is generally correct and clear.
The authors may consider the following comments for revising the paper:
- Paragraphs should not be separated by spaces.
- In Table 1 it is “g” not “gm”.
- The entire rehabilitation system is not presented. The positioning of the two linear motors does not result.
- Comments should be inserted regarding the compliant behavior of the rehabilitation system, meaning its adaptability to possible resistance put up by the patient upon the onset of pain. It is known that during rehabilitation exercising certain limits of the motions can be exceeded, beyond that the patient may feel pain. The response time of the rehabilitation equipment since the sensing of pain until the execution of corrections, that is until compliance is activated, should be as short as possible, this being an important feature of a recovery device.
Moderate editing of English language required
Round 2
Reviewer 1 Report
The authors have addressed almost all my suggestions, and the new version of the manuscript has been improved. However, there are some issues to correct. After addressing the following suggestions, the paper could be considered for publication in Sensors MDPI.
Specific comments:
* Once again, Table 6 is out of the format. Please verify and correct it.
* Regarding Figure 6. The title and axes' names are so big. Please standardize the Figures!
* The authors are advised to review the following paper, which can contribute to the state of the art:
Maldonado-Mejía et al., A fabric-based soft hand exoskeleton for assistance: the ExHand Exoskeleton, Frontiers in Neurorobotics, 2023, https://doi.org/10.3389/fnbot.2023.1091827
Reviewer 2 Report
- Paragraphs should not be separated by spaces.
- The entire rehabilitation system is not presented. The positioning of the two linear motors does not result.
Minor editing of English language required
